# Unveiling Transfer Learning Effectiveness Through Latent Feature Distributions

## Abstract

Transfer learning leverages large-scale pretraining to adapt models to specific downstream tasks. It has emerged as a powerful and widely adopted training strategy in deep learning frameworks. So, what makes it effective? Prior research has attributed its success to feature reuse, pretrained weights reuse, domain alignment, and the transfer of low-level data statistics. This study goes beyond these perspectives and focuses on a more fundamental factor: the evolution of logits distribution within the latent feature space of pretrained models. We introduce a novel approach using the Wasserstein distance to track distributional changes in the latent features. We find that pretraining not only learns the input distributions but also transforms them into generalizable internal representations in a consistent manner across all frozen layers. This finding underpins the effectiveness of transfer learning and provides a unifying explanation for these established theoretical perspectives.

## 1 Introduction

Transfer learning has emerged as a pivotal strategy in modern machine learning. It has driven some major breakthroughs in areas such as large language models (LLMs), computer vision, and multimodal systems. It works by first training a large, often overparameterized model on a broad dataset, then fine-tuning it for a more specific task. The idea is to reuse the learned knowledge, typically stored in the model's pretrained weights, from the original task (the source) to a different but related problem (the target). This approach is particularly helpful when the target task only has limited labeled data available, but the source task has plenty for training even a large model. In empirical applications, it is often applied to train a large model on massive general-purpose datasets and then adapt the pretrained model to a wide range of downstream domain-specific tasks with minimal additional training.

The successful adoption of transfer learning has brought renewed attention to two fundamental questions: (1) Does transfer learning truly improve performance across tasks? and (2) What underlying factors contribute to its effectiveness?

Regarding the first question, He et al. (2019a) argue that while pretraining accelerates convergence, it does not offer performance gains compared to standard training from scratch. Kornblith et al. (2019) observe that pretrained features are not universally transferable, particularly when the source and target tasks differ in distribution. In contrast, other studies (Hendrycks et al., 2019; 2020) show that pretraining yields significant performance and robustness boosts, especially when the target dataset contains label noise, adversarial perturbations, and out-of-distribution shifts, highlighting a more nuanced view of its effectiveness.

Regarding the second question, Raghu et al. (2020) find that feature reuse is the dominant factor in the success of meta-learning. In a separate study (Raghu et al., 2019), they further demonstrate that even reusing only the scaling of pre-trained weights, that is, only resampling samples with the same distribution parameters and without transferring the learned features, can lead to large gains in convergence speed. Neyshabur et al. (2020) identify three major sources of transferability: reuse of features, reuse of pretrained weights, and transfer of low-level data statistics. Furthermore, Mensink et al. (2021) underscore the importance of domain alignment, showing that positive transfer is most likely when the source domain sufficiently overlaps with the target domain. Domain adaptation methods have been extensively studied, and several effective

techniques have been proposed (Herath et al., 2019; Chen et al., 2019; Zhang et al., 2022; Li et al., 2021; Xu & Li, 2023).

In summary, the collective findings of these studies can be categorized into two groups:

1. Reuse of pre-trained weights (Raghu et al., 2019; Neyshabur et al., 2020; Parmar et al., 2024).

2. Data Distribution Learning, including:

   - Reuse of features (Raghu et al., 2019; Neyshabur et al., 2020; Li et al., 2024);
   - Transfer learning of low-level statistics in data (Neyshabur et al., 2020);
   - Domain alignment (Mensink et al., 2021; Teterwak et al., 2024).

   In our research scope, feature reuse and domain alignment can be interpreted as learning a shared data distribution across pretraining datasets.

Yet, a critical aspect remains unexplored: how pretraining influences the internal transformations of feature representations, particularly in terms of distributional dynamics throughout the model's latent space. Understanding these transformations could provide a unifying explanation for these existing theories about the generalization and transferability observed in pretrained models.

In this paper, we contribute to this understanding with the following findings:

1. We empirically explain why transfer learning is more effective than randomly initialized training, even though both can occasionally achieve similar prediction accuracy. Transfer learning demonstrates greater stability in training trajectories and final outcomes, whereas randomly initialized training is highly variable and may fail to converge due to unfavorable weights initialized in rare cases.

2. We demonstrate that the pretrained frozen layers produce the final representations z in a consistent manner, even across different input data. This allows the unfrozen layers to retain their learned functionality during pretraining to map the representations $z$ to the target with minor adjustments to the pretrained weights. This finding supports theories of weight reuse and basin convergence (Neyshabur et al., 2020).

3. We extend this transformation consistency by showing that it holds across the internal representations of all frozen layers, not just the final layer. Our perturbation experiments reveal that pretraining encourages the model to internalize the generalizable patterns across datasets while suppressing dataset-specific noise. These findings help explain the existing theories of feature reuse (Raghu et al., 2020) and domain alignment (Mensink et al., 2021).

4. We propose a novel method using Wasserstein distance to detect the evolution of logits distributions within the latent feature space of pretrained models, offering a new diagnostic lens for understanding the internal effects of pretraining on model behavior.

## 2 Methods

### 2.1 Data and dataset preparation

**Data source**. Based on the necessity for extensive pretraining across collections of datasets with common characteristics, we have determined that the stock market is an optimal data source. Prior studies have extensively explored the application of transfer learning to stock market prediction (He et al., 2019b; 2023; Pang et al., 2020). While each stock's trading records represent characteristics of an individual corporation, stocks within the same industry collectively contain some common industrial and market dynamics. This layered data structure is essential for testing how well transfer learning generalizes. Moreover, the high volatility, non-stationarity, and complex temporal dependencies of stock market data serve as a challenging, yet useful test for evaluating the reliability of our research. All data used in this study are publicly accessible on the Yahoo Finance website with the yfinance API.

**Data processing**. To facilitate effective pretraining, we download trading data of 937 stocks from the US finance industry at a daily frequency. This targeted selection is consistent with the findings of Mensink et al. (2021) that positive transfer gain occurs when the pretraining and target tasks share a domain. In addition, we download daily data for the Finance Index (Ticker: `^NYKTR`) and the S&P 500 (Ticker: `^GSPC`) at their maximum available length. All data are pre-processed as follows:

1. **Initial Trading Filter**: For each stock, early trading days are excluded if the daily trading volume continuously remained below 100 shares. This is to mitigate the irregular price movements and illiquidity effects commonly observed in the early stages following an initial public offering (IPO).

2. **Interpolation of Missing Values**: Missing values in the time-series are linearly interpolated to maintain continuity without introducing significant bias.

Next, we filter stocks with data availability from January 2, 2003, to August 2, 2024, a relatively recent market phase following the disruptions of the Dot-com bubble in 2002. This gives us a dataset of 336 stocks, each with 5,433 trading entries, adding up to over 1.8 million data points in total. The same data selection criteria are applied to the Finance Index and S&P 500 datasets.

**Feature selection**. All datasets of the 336 stocks contain five fields: Open, High, Low, Close, and Volume (OHLCV), which form the basis of features. We then supplement these 5 features of with the Close prices from the Finance Index and S&P 500. The target variable is defined as the one-day-ahead Close price relative to the features. For feature selection, we apply the Backward Elimination method, starting with all seven formulated features and iteratively removing the least significant ones. Our analysis reveals that the four OHLC (Open, High, Low, Close) features from the stocks retain the most predictive power for the target. This finding coincidentally aligns with the findings of Lakshminarayanan & McCrae (2019).

**Data split**. For LSTM training, we transform each of the 336 stock-specific datasets into window-framed datasets, using a window size of 7 and a shift step of 1. As a result, each training sample consists of 7 consecutive time steps of trading records, which are used to predict the close price of the next day. Notably, data samples are not shuffled within the datasets to preserve temporal dependencies at the window level.

For fine-tuning and testing and purposes, we first reserve 15 representative datasets (hereafter referred to as reserved datasets) based on stock price volatility, by selecting five from each volatility category: high, medium, and low. The remaining 321 datasets are used for pretraining (referred to as "pretraining datasets" later), with the training-validation-test split at 70/20/10 (test split are not used in pretraining).

## 2.2 Model architecture

**LSTM layer**. Numerous studies have investigated the optimal algorithm for stock market prediction, and many (Shah et al., 2018; Ma, 2020; Nabipour et al., 2020; Xiao et al., 2024) find that Long Short-Term Memory (LSTM) delivers superior performance. This advantage is attributed to its gated memory mechanism, which enables the retention of long-term historical information and enhances resilience to the vanishing gradient problem. The fundamental mechanisms of LSTM are demonstrated below:

$$\begin{aligned} \text{Forget gate:} \quad & f_t = \sigma\left(W_f[h_{t-1}; x_t] + b_f\right) \\ \text{Input gate:} \quad & i_t = \sigma\left(W_i[h_{t-1}; x_t] + b_i\right) \\ \text{Output gate:} \quad & o_t = \sigma\left(W_o[h_{t-1}; x_t] + b_o\right) \end{aligned}$$

The cell state and hidden state are computed as follows:

$$\begin{aligned} \text{Cell State:} \quad & C_t = f_t \otimes C_{t-1} + i_t \otimes \tanh\left(W_s[h_{t-1}; x_t] + b_s\right) \\ \text{Hidden State:} \quad & h_t = o_t \otimes \tanh(C_t) \end{aligned}$$

where $[h_{t-1}; x_t] \in \mathbb{R}^{m+n}$ denotes the concatenation of the previous hidden state $h_{t-1} \in \mathbb{R}^m$ and the current input $x_t \in \mathbb{R}^m$. $W_f, W_i, W_o, W_s \in \mathbb{R}^{m \times (m+n)}$ are the learnable weights, and $b_f, b_i, b_o, b_s \in \mathbb{R}^m$ are the

learnable bias terms. $C_{t-1}$ is the previous cell state. $\sigma$ denotes the logistic sigmoid function, and $\otimes$ represents element-wise (Hadamard) multiplication.

Furthermore, Qian & Chen (2019) reveal that LSTM exhibits a notable degree of robustness to data instability, further solidifying our selection of LSTM layer as the foundational building blocks of the predictive models, given this research uses numerous datasets that contain distributional variations in pretraining.

**Model construction**. Since the research objectives require performance testing under different experimental settings, full customization of the model structure is important for a fair comparison analysis of the experiment results. This requirement inspires us to build models from scratch using TensorFlow. Designing a well-functioning neural network involves careful consideration of model size, depth, and width, all of which should be appropriate for the problem complexity, dataset size, and computational constraints. The goal is to obtain strong predictive accuracy and robustness across datasets. We adhere to the following general principles in model construction.

1. Previous research (Neyshabur et al., 2017; Zhang et al., 2021; Choromanska et al., 2015) has claimed neural networks can generalize well even when the parameter size is larger than the sample size, provided that regularization is available to avoid overfitting. Therefore, we develop an overparameterized model and closely monitor overfitting in training.

2. Balancing the depth and width of the model is a key consideration in constructing an effective neural network. The empirical work (Safran & Shamir, 2017; Li et al., 2018) has reported that model depth has a more impact on model performance than model width. Besides, deep models can also benefit more from Transfer Learning (Bengio et al., 2011). Both insights direct this study to focus on model depth as a priority.

3. The primary source of generalization stems from the model structure itself (Zhang et al., 2021), while regularisation only helps prevent overfitting (Krogh & Hertz, 1991; Zhang et al., 2021). We opt not to incorporate regularizers, as they can ultimately degrade feature representations, even though they may offer slight performance gains during pre-training.

4. We do not use any normalization layers, considering that normalization could disrupt the temporal relationships underlying time-series data. As a compensation, we implement a small training batch size which serves as an implicit regularization to help gradients escape from sharp minima and promote better genalization (Keskar et al., 2016). As shown in Appendix A, the performance improvements from applying normalization are inconsistent and insignificant.

We construct 4 models of similar parameter size but varying numbers of LSTM layers (ranging from 6 to 8 layers), each with a different neuron distribution across the layers. Ultimately, the seven-layer LSTM model is selected as our baseline, considering its performance, robustness, and our research objectives. However, the remaining 3 candidates are retained in our experiments for comparison and analysis. The baseline model has 3 million parameters, compared to an input sample size of 1.8 million, and a maximum layer width of 512 neurons. The model structures of all 4 models are presented in Appendix B.

Finally, we select MAPE (Mean Absolute Percentage Error) as the loss function for all models, as it is a scale-independent metric. It satisfies our research requirement that the loss function should be agnostic to the price scale variations across different stocks.

$$\text{MAPE} = \frac{1}{n} \sum_{i=1}^{n} \left| \frac{y_i - \hat{y}_i}{y_i} \right| \times 100 \tag{1}$$

where $n$ is the number of data points; $y_i$ is the actual value for the $i^{\text{th}}$ observation; $\hat{y}_i$ is the predicted value for the $i^{\text{th}}$ observation.

### 2.3 Hypotheses setupe

To formulate our hypotheses, we begin with a probabilistic reasoning framework that models the input features $x$, target $y$, and latent representations $z$ in the feature space of the model.

Start with the joint probability rule:

$$p_{\text{train}}(y, x) = p(x)\, p_{\text{train}}(y \mid x) \tag{Equation 1}$$

Marginalize over latent variable $z$:

$$p_{\text{train}}(y \mid x) = \int p_{\text{train}}(y \mid z)\, p(z \mid x)\, dz \tag{Equation 2}$$

Replace Equation 2 in Equation 1:

$$p_{\text{train}}(y, x) = p(x) \int p_{\text{train}}(y \mid z)\, p(z \mid x)\, dz \tag{Equation 3}$$

For a given sample dataset, $p(x)$ is known. Therefore, we get:

$$p_{\text{train}}(y, x) \propto \int p_{\text{train}}(y \mid z)\, p(z \mid x)\, dz \tag{Equation 4}$$

Neural networks assume that the true input distribution $p(y, x)$ can be approximated from finite observable training samples $p_{\text{train}}(y, x)$ and that latent feature spaces can be learned to map the input distribution to useful internal representations $p(z \mid x)$ for predicting $p_{\text{train}}(y \mid x)$ with the mapping function $p_{\text{train}}(y \mid z)$ (Equation 2). Therefore, at the end, the training samples $p_{\text{train}}(y, x)$ are determined by the internal representations $p(z \mid x)$ and the mapping function $p_{\text{train}}(y \mid z)$ (Equation 4).

For the pretrain-then-tune paradigm, $p(z \mid x)$ represents the internal representations $z$ which are generated by frozen layers, while the mapping function $p_{\text{train}}(y \mid z)$ denotes the unfrozen layers where the internal representations $z$ are mapped to target $y$ with the reuse of pretrained weights. The effectiveness of pretraining, if it exists, should reside within these two components of the model. Therefore, we formulate our hypotheses as follows:

> **Hypothesis 1:** The pretrained weights in both frozen and unfrozen layers enable generalizable, stable and fast fine-tuning, enhancing the model's adaptability to new tasks and data.

> **Hypothesis 2:** Frozen layers in a pretrained model produce generalizable internal representations $z$ in a consistent manner before reaching to unfrozen layers, providing useful latent features for fine-tuning to effectively adapt the model to new tasks and data.

> **Hypothesis 3:** The effectiveness of transfer learning stems from the internal mechanism of pretrained models, which transforms inputs into a latent feature space in a consistent manner across all frozen hidden layers, enabling strong generalization across diverse tasks.

Hypothesis 1 aims to empirically validate the effectiveness of pretrained weights in fine-tuning compared to randomly initialized training. Hypothesis 2 and 3 explore the source of that effectiveness and thus provide fundamental supports to Hypothesis 1.

### 2.4 Effectiveness of pretrained weights

To test Hypothesis 1, we have designed and trained three closely related models to evaluate the effectiveness of pretrained weight in both frozen and unfrozen layers. A high level model configurations are summarized in Table 1.

We start with training and comparing One-layer-model and ft-Progressive. Since the difference between them lies in the presence of frozen layers which are missing in One-layer-model but included in ft-Progressive, any observed difference in predictive performance could be attributed to the pretrained weights in the frozen layers.

Next, we focus on training and comparing ft-Progressive and ft-Unfrozen, which share identical frozen layers but differ in how the model head is initialized. Since the frozen layers are fixed and always output deterministic internal representations wihch can be seen as transformed inputs to the model head, thus ft-Unfrozen represents fine-tunning leveraging pretrained weights, whereas ft-Progressive functions more like standard training from scratch. Thus, any performance difference could be attributed to the reuse of pretrained weights in the unfrozen layer, the model head in this case.

We utilized two metrics obtained from the fine-tuning process to assess the effectiveness of transfer learning.

1. The stability of loss convergence. A stable and smooth convergence curve suggests that the trainable model head is initialized with well-informed weights, allowing the model to map the fixed latent representations z to the target outputs without erratic fluctuations during fine-tuning.

2. The L2 (Euclidean) distance in model head's parameter space across runs. A smaller distance suggests that the pretrained model head has already encoded useful and generalizable information, requiring only minimal weight adjustments when adapted to different downstream tasks.

Table 1: Model configurations. All models only have one trainable layer, the model head. The difference between One-layer-model and *ft-Progressive* lies in the presence of frozen layers which are missing in One-layer-model. The distinction between *ft-Unfrozen* and *ft-Progressive* is in the model head, where *ft-Unfrozen* reuses pretrained weights, while *ft-Progressive* uses random initialization.

| Models | Hidden Layers | Fine-tuning Method | Model Head |
|---|---|---|---|
| One-layer-model | None | None | Randomly initialized weights |
| ft-Progressive | Frozen Baseline | Replace with a new head | Randomly initialized weights |
| ft-Unfrozen | Frozen Baseline | Unfreeze model head | Pretrained weights reuse |

## 2.5 Logits distribution evolution

To test Hypothesis 2 & 3, we investigate the distributions of logits in the latent feature space of the model. However, those distributions are unknown and difficult to estimate and interpret explicitly, since they are dynamically shaped by the network architecture, activation functions, and optimization process during training. Instead, we use the Wasserstein distance to quantify distributional differences in the input space (between two subsets derived from a larger dataset) and in the latent feature space (between their corresponding logits at each model layer). The concept of logits distance, reflecting the degree of distributional difference, is illustrated in Figure 1. By tracking these distance changes, we gain insights into the layer-by-layer evolution of logits distributions within the latent feature spaces during training. This, in turn, helps us understand why transfer learning is effective in enabling a model to adapt from the pretrained tasks and data to new ones.

The Wasserstein distance quantifies the minimum "work" needed to transform one distribution into another by moving "mass" around, where the "work" is defined by the distance and the amount that the mass is moved. It is particularly well-suited for our research because it does not require explicit knowledge of distributions being compared. This is critically important because the distribution of internal representations is typically unknown, and applying a density estimation function, such as the commonly used Kernel Density Estimation (KDE), could lead to information loss.

In this study, we use the Wasserstein-1 distance, or Earth Mover's distance, defined as:

$$W_1(P, Q) = \inf_{\gamma \in \Gamma(P,Q)} \int_{\mathcal{X} \times \mathcal{X}} d(x, y) \, d\gamma(x, y) \qquad (2)$$

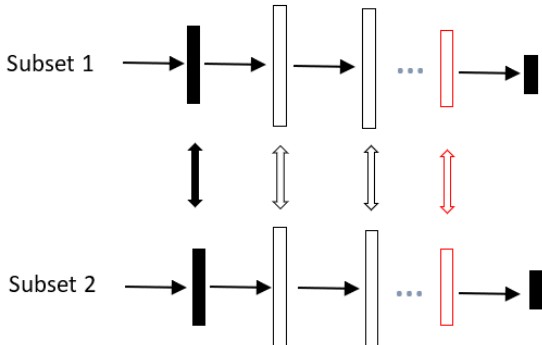

Figure 1: Concept of distance. The two subsets are split from a larger dataset at 70/30. When they pass through the same model, their output logits from each frozen layer are saved, and the Wasserstein-1 distances are calculated between their logits from the same layer.The black filled blocks represent input and output layers. The red framed blocks represent the last forzen layer which ouput the internal representation z as an input to unfrozen model head (the output layer) for fine-tuning.

where $\Gamma(P, Q)$ is the set of all couplings (joint distributions) of $P$ and $Q$ that have marginals $P$ and $Q$ (i.e., $\gamma(\mathbf{x}) = P(\mathbf{x})$ and $\gamma(\mathbf{y}) = Q(\mathbf{y})$). $d(\mathbf{x}, \mathbf{y})$ is the distance between points $x$ and $y$ in the metric space $\mathcal{X}$, which is defined by the Euclidean distance in this study. The infimum, denoted as inf, is taken over all possible couplings $\gamma$.

The codes of the Wasserstein distance function are enclosed in Appendix C.

## 3  Experiments & results

### 3.1  Effectiveness of pretrained weights

This experiment tests Hypothesis 1: whether pretrained weights enable generalizable, stable and fast fine-tuning, enhancing the model's adaptability to new tasks and data.

**Experiment**. We first pretrain the 7-LSTM-layer baseline model with the parameters specified in Table 7, using the 321 pretraining datasets. The optimal training instances of model, saved when the validation loss is at its lowest during pretraining, are then used as the base model for training on the three models described in Section 2.4, with the parameters specified in Table 7.

Each of the three models are trained with the same 15 reserved datasets, split at 70/20/10 for training, validation, and evaluation. The experiment results demonstrate consistency among all of these datasets. For presentation purposes, we only show results from five representative datasets [CET, CFFN, VABK, BUSE, HGBL], which are selected at every 3rd step along the volatility ranking of the 15 reserved datasets.

The training is performed on each dataset separately using the optimal learning rate determined through grid search. Once the learning rate is set, fine-tuning is repeated multiple times per dataset. For each run, we save the trainable weights, the training loss per epoch, and the evaluation loss, which are subsequently used for analysis as follows:

- Trainable weights: used to calculate the L2 distances in the parameter space across runs.

- Training loss: used to analyze the training trajectory over repeated runs with the same dataset.

- Evaluation loss: used to assess fine-tuning performance.

**Results & discussion**. Table 2 presents the performance comparison between ft-Progressive and the One-Layer-Model, which differ in the presence of frozen layers while sharing an identical final trainable layer, as

detailed in Section 2.4. The substantial outperformance of ft-Progressive (marked in red font) suggests that the frozen pretrained layers have already transformed the inputs into meaningful internal representations by the time they reach the model head. This significant performance gain indicates that the pretrained weights in the frozen layers are primarily responsible for the improvement, a hypothesis that will be further investigated in Section 3.2.

The evaluation results showing in Table 3 presents that random initialization (ft-Progressive) can achieve competitive predictions and even occasionally outperform pretrained fine-tuning (ft-Unfrozen), as marked in red font in the table. This observation is consistent with the findings of He et al. (2019). However, ft-Progressive exhibits high volatility across runs, whereas ft-Unfrozen demonstrates much more stable performance. For certain perturbed datasets, such as HGBL, ft-Progressive struggles to converge on a good minima, while ft-Unfrozen consistently achieves superior results. This finding aligns with the robustness benefits of pretraining reported by Hendrycks et al. (2019, 2020). The effectiveness of pretrained fine-tuning is further reinforced by the following observations:

1. Table 4 shows that ft-Unfrozen yields consistently low (near-zero) L2 distances in parameter space across datasets, and that these results remain stable across runs. In contrast, ft-Progressive produces larger and more differing L2 distances both across datasets and across runs. These observations indicate that reusing pretrained weights reduce the need for significant parameter updates in fine-tuning, effectively guiding optimization toward the same basin of minima across different datasets. This finding is in line with the work of Neyshabur et al. (2020).

2. Table 5 & 6 reinforces the findings from Point 1. ft-Progressive produces variable L2 distances across runs on the same dataset (Table 5), while ft-Unfrozen yields zero distance (Table 6). This is because in ft-Unfrozen, both the frozen representations z and the pretrained model head remain fixed, leaving the input dataset as the only source of variation. Thus, the identical input should yield consistent fine-tuning outcomes.

3. Figure 2 shows that ft-Unfrozen produces the same training trajectories across repeated runs on the same dataset, while ft-Progressive results in different trajectories. This contrast is due to how the models are initialized: ft-Unfrozen starts with pretrained weights, whereas ft-Progressive is randomly initialized. Furthermore, ft-Unfrozen is observed to converge in fewer epochs, indicating faster and more efficient training, another clear benefit of reusing pretrained weights.

This experiment supports Hypothesis 1, showing that pretrained weights in both frozen and unfrozen layers promote generalizable, stable, and efficient fine-tuning, thereby improving the model's adaptability to new tasks. However, the underlying mechanisms behind these benefits remain unclear and will be further investigated in Section 3.2 and 3.3.

Table 2: Performance of pretrained weights in frozen layers. One-layer-model shares identical structure with the last layer of *ft-Progressive* (randomly initialized). However, it does not have the frozen layers. The results of *ft-Progressive* (lower MAPE marked in red font) overwhelmingly overperform those of One-layer-model by large margins.

| **ft-Progressive** | | | | | | **One-layer-model** | | | | |
| **ticker** | Run1 | Run2 | Run3 | Run4 | Run5 | **ticker** | Run1 | Run2 | Run3 | Run4 | Run5 |
|---|---|---|---|---|---|---|---|---|---|---|---|
| CET | 2.91 | 0.93 | 0.89 | 0.74 | 0.69 | CET | 3.31 | 3.43 | 3.39 | 3.38 | 3.27 |
| CFFN | 2.37 | 1.68 | 4.07 | 4.37 | 1.66 | CFFN | 9.04 | 9.67 | 8.80 | 9.85 | 8.99 |
| VABK | 1.63 | 1.62 | 1.50 | 2.34 | 1.51 | VABK | 8.61 | 8.51 | 8.88 | 8.52 | 8.48 |
| BUSE | 1.37 | 1.47 | 1.38 | 1.39 | 1.38 | BUSE | 6.94 | 6.90 | 7.06 | 6.87 | 6.87 |
| HGBL | 26.57 | 62.64 | 35.57 | 28.10 | 30.07 | HGBL | 23.25 | 27.45 | 28.06 | 40.17 | 27.78 |

Table 3: Performance of pretrained weights in unfrozen layers. *ft-Unfrozen* (unfrozen) results remain consistent across all runs, while *ft-Progressive* (randomly initialized) varies due to random weights initiation. Although *ft-Progressive* can occasionally produce better results (lower MAPE marked in red font), this is not guaranteed. For some datasets, such as HGBL, *ft-Progressive* has difficulties converging to a good minima.

**ft-Progressive**

| ticker | Run1 | Run2 | Run3 | Run4 | Run5 |
|--------|------|------|------|------|------|
| CET | 2.91 | 0.93 | 0.89 | 0.74 | 0.69 |
| CFFN | 2.37 | 1.68 | 4.07 | 4.37 | 1.66 |
| VABK | 1.63 | 1.62 | 1.50 | 2.34 | 1.51 |
| BUSE | 1.37 | 1.47 | 1.38 | 1.39 | 1.38 |
| HGBL | 26.57 | 62.64 | 35.57 | 28.10 | 30.07 |

**ft-Unfrozen**

| ticker | Run1 | Run2 | Run3 | Run4 | Run5 |
|--------|------|------|------|------|------|
| CET | 0.69 | 0.69 | 0.69 | 0.69 | 0.69 |
| CFFN | 2.00 | 2.00 | 2.00 | 2.00 | 2.00 |
| VABK | 1.53 | 1.53 | 1.53 | 1.53 | 1.53 |
| BUSE | 1.43 | 1.43 | 1.43 | 1.43 | 1.43 |
| HGBL | 5.08 | 5.08 | 5.08 | 5.08 | 5.08 |

Table 4: L2 distances of weights across datasets. The L2 distances are calculated with weights obtained from using different datasets under the same run. The tables are organized by runs, with both row and column indices representing stock tickers (datasets). *ft-Unfrozen* (unfrozen model head) produces identical L2 distances across runs (Run1 vs. Run2), while *ft-Progressive* (randomly initialized model head) yields varying L2 distances across runs. Moreover, the distances under *ft-Unfrozen* are significantly lower (near zero), highlighting the stability and consistency afforded by pre-trained weights.

**ft-Unfrozen**

**Run1**

| | CET | CFFN | VABK | BUSE | HGBL |
|------|-----|------|------|------|------|
| CET | – | 0.00 | 0.00 | 0.01 | 0.24 |
| CFFN | – | – | 0.00 | 0.01 | 0.24 |
| VABK | – | – | – | 0.01 | 0.24 |
| BUSE | – | – | – | – | 0.23 |
| HGBL | – | – | – | – | – |

**Run2**

| | CET | CFFN | VABK | BUSE | HGBL |
|------|-----|------|------|------|------|
| CET | - | 0.00 | 0.00 | 0.01 | 0.24 |
| CFFN | | - | 0.00 | 0.01 | 0.24 |
| VABK | | | - | 0.01 | 0.24 |
| BUSE | | | | - | 0.23 |
| HGBL | | | | | - |

**ft-Progressive**

**Run1**

| | CET | CFFN | VABK | BUSE | HGBL |
|------|-----|------|------|------|------|
| CET | - | 3.08 | 3.13 | 2.37 | 11.17 |
| CFFN | | - | 2.37 | 2.40 | 11.24 |
| VABK | | | - | 2.33 | 10.36 |
| BUSE | | | | - | 11.31 |
| HGBL | | | | | - |

**Run2**

| | CET | CFFN | VABK | BUSE | HGBL |
|------|-----|------|------|------|------|
| CET | - | 2.24 | 1.74 | 2.54 | 6.45 |
| CFFN | | - | 1.88 | 2.68 | 5.76 |
| VABK | | | - | 2.60 | 6.65 |
| BUSE | | | | - | 6.57 |
| HGBL | | | | | - |

## 3.2 Consistent transformation in the last frozen layer

This experiment tests Hypothesis 2: whether the frozen layers can generate the final layer representations z in a consistent manner, enabling fine-tuning to effectively adapt to new tasks and data.

**Experiment**. To obtain an unbiased understanding about the final frozen layer representations z, we train all four pretraining models detailed in Section 2.2, using the same datasets, methods, and parameter settings as shown in Table 7. For each of the four models, we save three training statuses, resulting in a total of 12 training instances:

- models $\in$ [Baseline, DiffNeuron[1], LessLayer[2], MoreLayer[3]]

---

[1] 7 LSTM layers, same as baseline model, but different neurons distribution among LSTM layers, with the total parameter size remaining approximately the same.

[2] 6 LSTM layers (vs 7 layers in baseline model), different neurons distribution among LSTM layers, with the total parameter size remaining approximately the same.

[3] 8 LSTM layers (vs 7 layers in baseline model), different neurons distribution among LSTM layers, with the total parameter size remaining approximately the same.

Table 5: L2 Distances of weights using *ft_Progressive* method. The L2 distances are calculated between runs based on the same input dataset. The tables are organized by stock tickers (datasets), with columns and rows representing repeated runs. *ft_Progressive* (randomly initialized model head) results in non-zero and varying L2 distances between runs, indicating unstable weight updates due to random initialization.

### ft-Progressive

| CET | Run1 | Run2 | Run3 | Run4 | Run5 |
|------|------|------|------|------|------|
| Run1 | - | 2.75 | 2.75 | 2.47 | 2.91 |
| Run2 | | - | 2.13 | 1.80 | 2.04 |
| Run3 | | | - | 1.57 | 1.76 |
| Run4 | | | | - | 1.85 |
| Run5 | | | | | - |

| CFFN | Run1 | Run2 | Run3 | Run4 | Run5 |
|------|------|------|------|------|------|
| Run1 | - | 2.26 | 2.70 | 2.65 | 2.19 |
| Run2 | | - | 2.57 | 2.38 | 2.27 |
| Run3 | | | - | 2.47 | 1.90 |
| Run4 | | | | - | 2.23 |
| Run5 | | | | | - |

| BUSE | Run1 | Run2 | Run3 | Run4 | Run5 |
|------|------|------|------|------|------|
| Run1 | - | 2.56 | 2.07 | 1.90 | 1.97 |
| Run2 | | - | 2.36 | 2.58 | 2.43 |
| Run3 | | | - | 2.37 | 2.00 |
| Run4 | | | | - | 1.88 |
| Run5 | | | | | - |

| VABK | Run1 | Run2 | Run3 | Run4 | Run5 |
|------|------|------|------|------|------|
| Run1 | - | 2.15 | 2.32 | 2.52 | 2.35 |
| Run2 | | - | 1.60 | 2.24 | 2.05 |
| Run3 | | | - | 2.00 | 1.82 |
| Run4 | | | | - | 2.30 |
| Run5 | | | | | - |

Table 6: L2 Distances of weights using *ft*-Unfrozen method. The L2 distances are calculated across runs based on the same input dataset. The results consistently show zero distances, indicating that fine-tuning with the *ft*-Unfrozen method (unfrozen model head) results in identical weights across repeating runs, demonstrating deterministic and stable training behavior.

### ft-Unfrozen

| Runs | Run1 | Run2 | Run3 | Run4 | Run5 |
|------|------|------|------|------|------|
| Run1 | - | - | - | - | - |
| Run2 | | - | - | - | - |
| Run3 | | | - | - | - |
| Run4 | | | | - | - |
| Run5 | | | | | - |

- status $\in$ [No-training[4], Optimal[5], Fully-trained[6]]

We feed each of the 12 training instances with two subsets, created by a 70/30 split (without shuffling) from each of the 15 reserved datasets. For each run, we extract the logits from the last frozen layer and compute the Wasserstein distance between the logits obtained from the two subsets on the same layer. The results are presented in Appendix D. Furthermore, to validate the generalizability of our findings, we repeat the same experiment but with shuffled input data. The results are shown in Appendix E. Their summary for analysis and discussion is shown in Figures 3 and 4.

**Results & discussion**. As shown in Appendix D and E, the raw experiment data in the first tables exhibits significant variation across datasets due to price scale differences among stocks. To mitigate this, we normalize each distance by dividing it by its corresponding input data distance (from the Input_Data

---

[4]the model instance saved with randomly initialized weights, without any training applied.
[5]the model instance saved with weights when the validation loss was at its lowest during pretraining.
[6]the model instance saved with fully trained weights after the pretraining is completed with all 321 pretraining datasets.

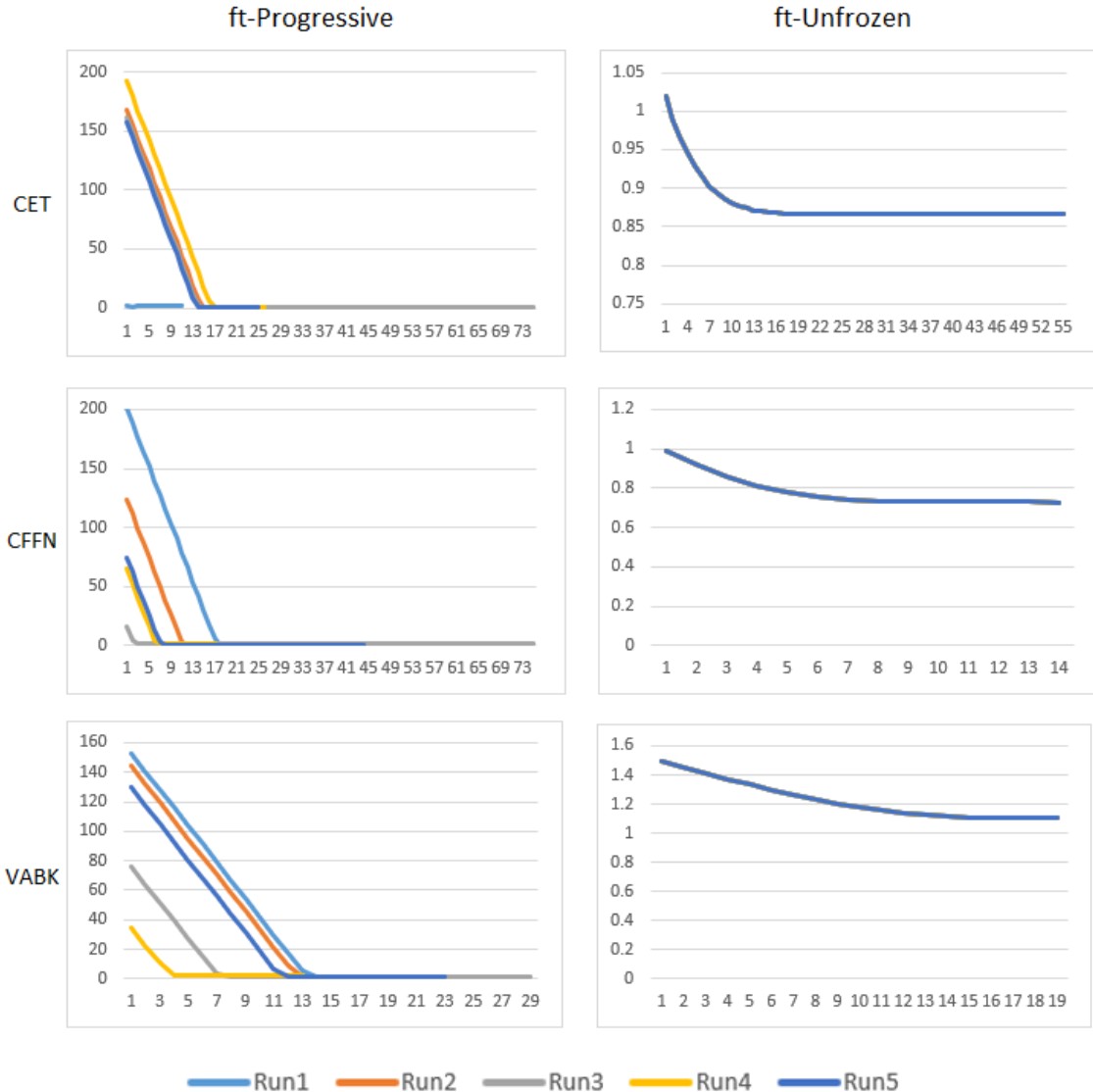

Figure 2: Training loss trajectory. The graph is organized by stock tickers (datasets). Under *ft_Unfrozen* (unfrozen model head), all training trajectories overlaps into a single line due to identical training path across runs. In contrast, *ft_Progressive* (randomly initialized model head) displays diverse training paths, reflecting variability caused by different randomly initialized weights in each run.

column), which results in the second tables. This normalization reveals significantly reduced variation in column-wise values across the 15 stocks.

To quantify the dispersion of column-wise values, we compute the coefficient of variation (CV), defined as the ratio of standard deviation to mean. Lower CV values indicate reduced dispersion and greater stability in distance distributions. Across the Fully-trained and Optimal models, CVs are consistently low for both non-shuffled (Figure 3; 0.08-0.13) and shuffled (Figure 4; 0.06–0.24) inputs. In contrast, No-training models show much higher variability, with CVs ranging from 0.30 to 0.48 for non-shuffled (Figure 3) and from 0.45 to 0.78 for shuffled (Figure 4).

These results suggest that the pretrained frozen layers not only capture the underlying input distribution, but also produce consistent final representations z across different inputs. This consistency is crucial because it ensures the generalizability of the the representations z which is an input to the unfrozen model head,

largely preserving the learned functionality stored in the pretrained weights to map representations z to the target. As a result, only minor weight adjustments are needed during fine-tuning, reducing the risk of large gradient updates which could escape the original basin of minima.

Although the results of this experiment strongly support Hypothesis 2, the internal mechanism of the model has not yet been examined, which will be addressed in Section 3.3.

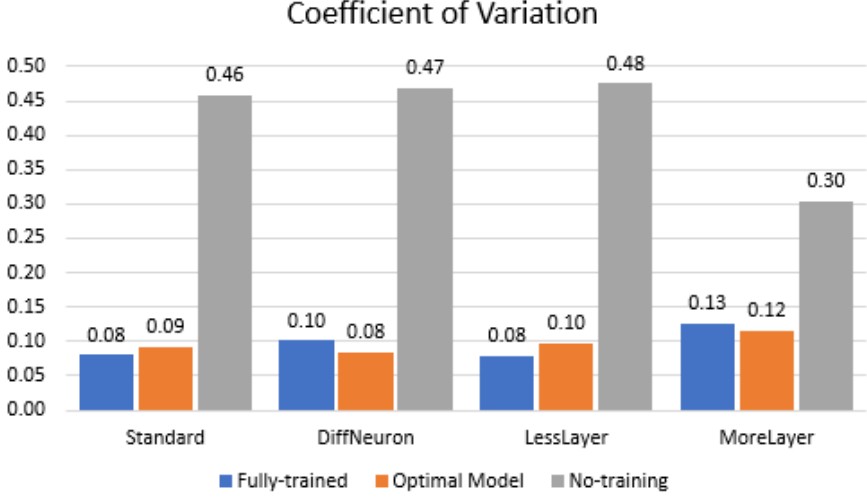

Figure 3: CVs from non-shuffled input data. Compare CVs (Coefficient of Variations) obtained with non-shuffled inputs for different model configurations (Standard, DiffNeuron, LessLayer, MoreLayer), each evaluated under three training conditions: Fully-trained, Optimal Model, and No-training.

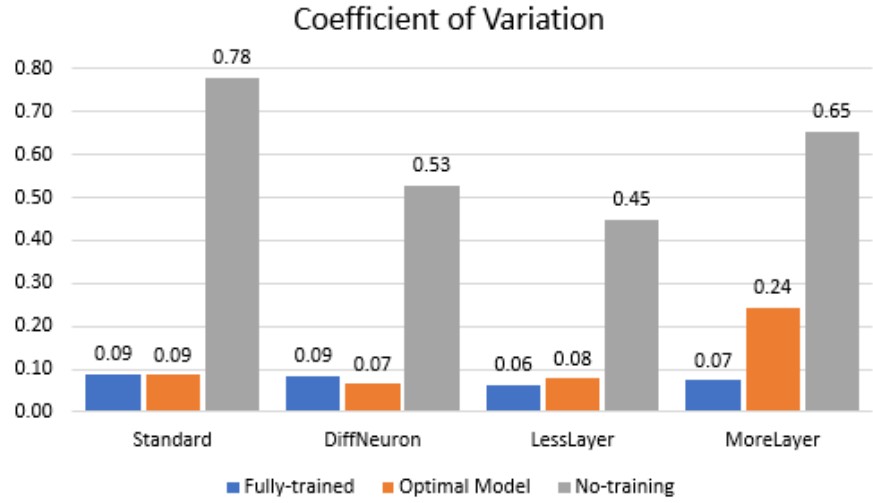

Figure 4: CVs from shuffled input data. Compare CVs (Coefficient of Variations) obtained with shuffled inputs for different model configurations (Standard, DiffNeuron, LessLayer, MoreLayer), each evaluated under three training conditions: Fully-trained, Optimal Model, and No-training.

### 3.3 Consistent transformation across all frozen layers

This experiment tests Hypothesis 3: examine how the evolution of logit distributions within pretrained models impacts transfer learning.

Table 7: Summary of training Parameters. The optimal learning rates of *ft_Progressive* (randomly initialized model head), *ft_Unfrozen* (unfrozen model head), and One-layer-model are determined through grid search during model training.

| Hyper-parameter | Pretraining | *ft_Progressive* | *ft_Unfrozen* | One-layer-model |
|---|---|---|---|---|
| optimizer | Adam | Adam | Adam | Adam |
| activation (LSTM) | leaky_relu | N/A | N/A | N/A |
| activation (Dense) | linear | linear | linear | linear |
| kernel_initializer (LSTM) | he_normal | N/A | N/A | N/A |
| kernel_initializer (Dense) | glorot_uniform | glorot_uniform | glorot_uniform | glorot_uniform |
| learning_rate | 0.0005 | varies | varies | varies |
| loss_func | MAPE | MAPE | MAPE | MAPE |
| batch_size | 100 | 100 | 100 | 100 |
| epochs | 200 | 300 | 300 | 300 |
| convergence | Earlystopping | Earlystopping | Earlystopping | Earlystopping |
| patience | 10 | 10 | 10 | 10 |
| monitor | MAPE | MAPE | MAPE | MAPE |

**Experiment**. This experiment is based on the experiment in Section 3.2, with two key modifications:

1. Instead of extracting logits only from the final frozen layer, we collect logits from all frozen layers to observe how the distributional distance evolves across the network.

2. We add two new datasets for perturbation testing: the S&P 500 dataset, which shares market dynamics with the 15 reserved stock datasets, and a synthetic dataset which is generated from a uniform distribution over [1, 100].

As in the experiment in Section 3.2, the distances are first computed from the raw logits, then normalized to remove price scale effects. For presentation purposes, Figure 5 only shows results from 5 representative datasets, selected using the same method as in the experiment in Section 3.1. The remaining datasets exhibit similar trends and support the same conclusions. The degree of line clustering in Figure 5 reflects the coefficient of variation (CV), both of which measure distributional consistency and dispersion across layers.

**Results & discussion**. The most notable pattern in Figure 5 is that the stock dataset lines cluster tightly together in both in the Fully-trained and Optimal models, while they appear scattered in the No-training models. This indicates that well-pretrained models transform input distributions across all frozen layers, not just the final one (in Experiment 2), in a consistent manner. This supports the common fine-tuning practice of unfreezing multiple top layers while keep the remaining frozen, depending on specific-task needs, since internal representations at each layer reflect consistent, shared transformations of the input data.

Another key observation is that the S&P 500 line clusters closely with the stock dataset lines, while the Random dataset line diverges significantly. This suggests that the S&P 500 shares underlying distributional patterns with individual stocks, likely because it reflects broad market dynamics that are inherently embedded in the stock prices. In contrast, the random dataset lacks any shared statistical pattern. This finding suggests that during pretraining, the model appears to internalize generalizable distribution patterns across datasets while filtering out the patterns that are specific to individual dataset, within the limits of model capacity. This insight supports the theoretical perspectives of feature reuse (Raghu et al., 2020; Neyshabur et al., 2020), transfer of low-level data statistics (Neyshabur et al., 2020), and domain alignment (Mensink et al., 2021), all of which emphasize the learning of shared distribution patterns during pretraining.

These results support Hypothesis 3, which posits that the internal mechanisms of the model are the source of transfer learning effectiveness. The experiment shows that pretraining transforms input distributions

into latent representations in a consistent manner across all frozen layers, a consistency that underpins the model's adaptability to new data and tasks.

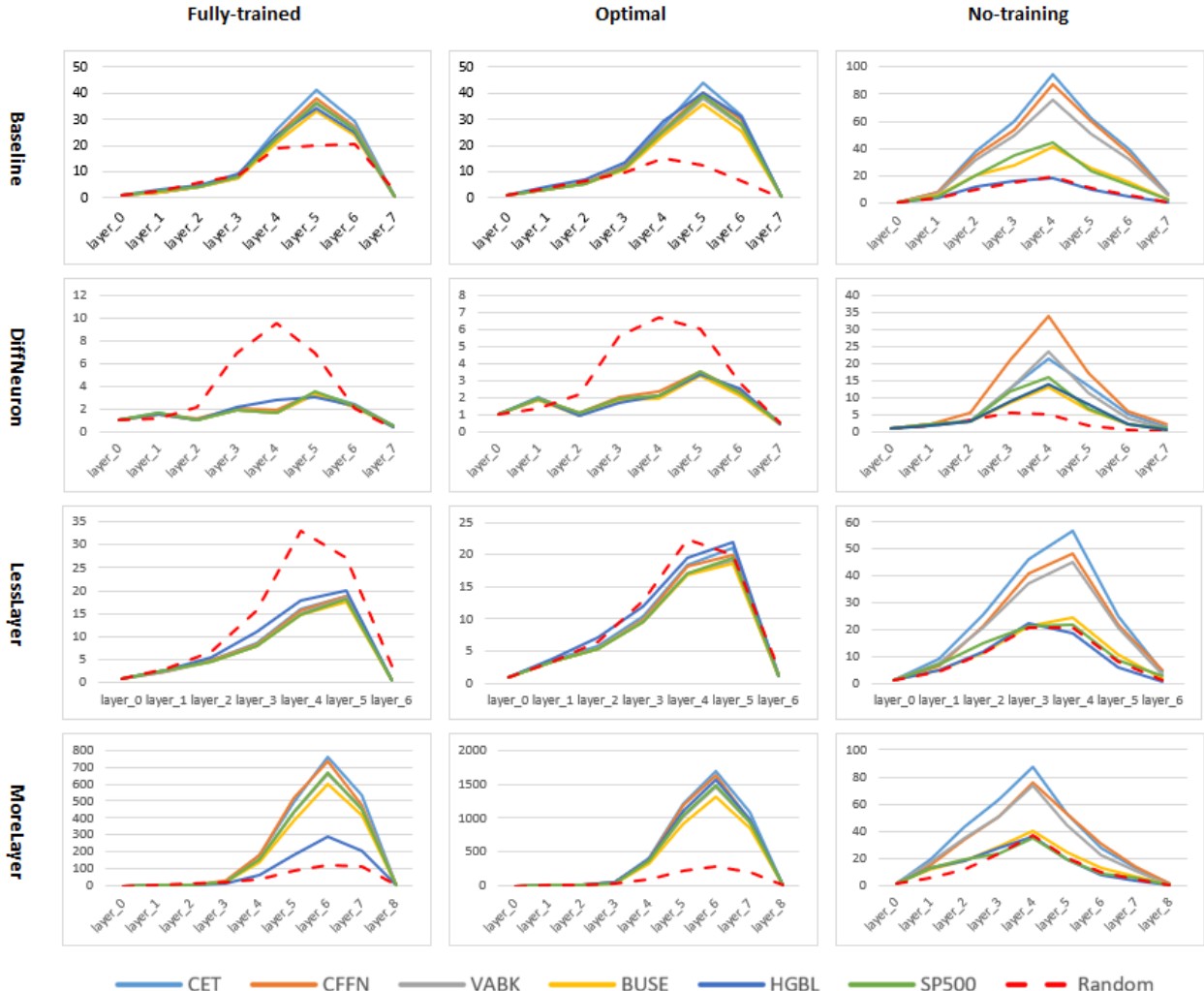

Figure 5: Distances by layers. The graphs are organized by model (row-wise) and training status (column-wise). Each line in the graph corresponds to the distances obtained from a specific dataset. The stock lines cluster closely together in Fully trained and Optimal models, while they are randomly dispersed in No-training models.

## 4 Conclusion

This study first empirically validates the effectiveness of pretrained weights in pretrained models, and then investigates the underlying source of this effectiveness by tracking distributional transformations within the latent feature space. To achieve this, we propose a novel approach that uses the Wasserstein distance to quantify distributional differences in logits obtained from the same frozen layers between two input subsets which are split from the same dataset. Our results demonstrate that a pretrained model transforms the inputs into generalizable internal representations in a consistent manner throughout all frozen layers. This transformation consistency underpins the accuracy, robustness, and generalizability of transfer learning. In addition, reusing pretrained weights in unfrozen layers further enhances the stability and convergence of the fine-tuning process.

Our findings offer a unifying explanation for several established perspectives in transfer learning, including feature reuse, pretrained weight transfer, domain alignment, and the propagation of low-level data statistics (Raghu et al., 2020; Neyshabur et al., 2020; Mensink et al., 2021). By showing that consistent latent transformations drive generalization, this study conceptually bridges these theories under a common framework.

Finally, our findings highlight the value of Wasserstein distance as a powerful tool in representation learning, suggesting its promise for deeper analysis. We plan to extend our future research in this direction.

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

## Appendix A: Standard vs Normalization.

Standard training in this experiment does not contain Normalization layer. From the Delta(A-B), it can be observed that the performance improvements from applying normalization are insignificant and inconsistent in our experiments.

**MAPE Loss (%)**

| Ticker | Volatility | Standard (A) | Normalization (B) | Delta (A-B) |
|--------|-----------|--------------|-------------------|-------------|
| CET | 1.2% | 0.95 | 1.35 | (0.40) |
| AJG | 1.4% | 2.30 | 10.03 | (7.73) |
| FNWD | 1.5% | 1.85 | 1.74 | 0.11 |
| CFFN | 1.5% | 8.19 | 4.60 | 3.58 |
| RY | 1.5% | 1.21 | 1.29 | (0.08) |
| FDBC | 2.4% | 2.35 | 1.93 | 0.42 |
| VABK | 2.4% | 1.64 | 1.64 | 0.00 |
| FCF | 2.4% | 1.53 | 1.50 | 0.03 |
| PFC | 2.4% | 1.85 | 1.78 | 0.07 |
| BUSE | 2.4% | 1.39 | 1.60 | (0.21) |
| SLNH | 8.8% | 9.77 | 7.66 | 2.11 |
| CARV | 9.0% | 5.92 | 4.25 | 1.67 |
| HGBL | 13.9% | 4.74 | 4.79 | (0.05) |
| KINS | 16.5% | 5.25 | 4.03 | 1.21 |
| WT | 49.8% | 1.75 | 1.39 | 0.35 |
| **mean** | 7.8% | 3.38 | 3.31 | 0.07 |
| **std** | 12.6% | 2.76 | 2.63 | 2.42 |
| **min** | 1.2% | 0.95 | 1.29 | (7.73) |
| **25%** | 1.5% | 1.59 | 1.55 | (0.07) |
| **50%** | 2.4% | 1.85 | 1.78 | 0.07 |
| **75%** | 8.9% | 4.99 | 4.43 | 0.82 |
| **max** | 49.8% | 9.77 | 10.03 | 3.58 |

# Appendix B: Transfer learning model structures

The structures of the 4 pretraining models.

1. Baseline

| Layer (type) | Output Shape | Param # |
|---|---|---|
| input_layer (InputLayer) | (None, 7, 4) | 0 |
| lstm (LSTM) | (None, 7, 32) | 4,736 |
| lstm_1 (LSTM) | (None, 7, 128) | 82,432 |
| lstm_2 (LSTM) | (None, 7, 256) | 394,240 |
| lstm_3 (LSTM) | (None, 7, 512) | 1,574,912 |
| lstm_4 (LSTM) | (None, 7, 256) | 787,456 |
| lstm_5 (LSTM) | (None, 7, 128) | 197,120 |
| lstm_6 (LSTM) | (None, 32) | 20,608 |
| dense (Dense) | (None, 1) | 33 |

Total params: 3,061,537 (11.68 MB)
Trainable params: 3,061,537 (11.68 MB)
Non-trainable params: 0 (0.00 B)

2. Different Neuron Distributions (DiffNeuron)

| Layer (type) | Output Shape | Param # |
|---|---|---|
| input_layer (InputLayer) | (None, 7, 4) | 0 |
| lstm (LSTM) | (None, 7, 32) | 4,736 |
| lstm_1 (LSTM) | (None, 7, 64) | 24,832 |
| lstm_2 (LSTM) | (None, 7, 256) | 328,704 |
| lstm_3 (LSTM) | (None, 7, 551) | 1,780,832 |
| lstm_4 (LSTM) | (None, 7, 256) | 827,392 |
| lstm_5 (LSTM) | (None, 7, 64) | 82,176 |
| lstm_6 (LSTM) | (None, 32) | 12,416 |
| dense (Dense) | (None, 1) | 33 |

Total params: 3,061,121 (11.68 MB)
Trainable params: 3,061,121 (11.68 MB)
Non-trainable params: 0 (0.00 B)

3. Less Layers (LessLayer)

| Layer (type) | Output Shape | Param # |
|---|---|---|
| input_layer (InputLayer) | (None, 7, 4) | 0 |
| lstm (LSTM) | (None, 7, 32) | 4,736 |
| lstm_1 (LSTM) | (None, 7, 128) | 82,432 |
| lstm_2 (LSTM) | (None, 7, 384) | 787,968 |
| lstm_3 (LSTM) | (None, 7, 512) | 1,837,056 |
| lstm_4 (LSTM) | (None, 7, 128) | 328,192 |
| lstm_5 (LSTM) | (None, 32) | 20,608 |
| dense (Dense) | (None, 1) | 33 |

Total params: 3,061,025 (11.68 MB)
Trainable params: 3,061,025 (11.68 MB)
Non-trainable params: 0 (0.00 B)

4. More Layers (MoreLayer)

| Layer (type) | Output Shape | Param # |
|---|---|---|
| input_layer (InputLayer) | (None, 7, 4) | 0 |
| lstm (LSTM) | (None, 7, 64) | 17,664 |
| lstm_1 (LSTM) | (None, 7, 128) | 98,816 |
| lstm_2 (LSTM) | (None, 7, 256) | 394,240 |
| lstm_3 (LSTM) | (None, 7, 502) | 1,524,072 |
| lstm_4 (LSTM) | (None, 7, 256) | 777,216 |
| lstm_5 (LSTM) | (None, 7, 128) | 197,120 |
| lstm_6 (LSTM) | (None, 7, 64) | 49,408 |
| lstm_7 (LSTM) | (None, 10) | 3,000 |
| dense (Dense) | (None, 1) | 11 |

Total params: 3,061,547 (11.68 MB)

Trainable params: 3,061,547 (11.68 MB)

Non-trainable params: 0 (0.00 B)

## Appendix C: Wasserstein distance

Uses matrix slicing method to measure pair-wise Euclidean distance for matrix (Frobenius Distance). It applies to both 2D array and 3D arrays, producing the same results if matching elements are the same.

```python
import numpy as np
import ot

def wasserstein_distance(data1, data2):
    # Generate uniform weights for each point (probabilities sum to 1)
    weights1 = np.ones((data1.shape[0],)) / data1.shape[0]
    weights2 = np.ones((data2.shape[0],)) / data2.shape[0]

    distances = np.zeros((data1.shape[0], data2.shape[0]))

    # Compute distances between corresponding 2D matrices (slices) in X and Y
    # keep 3D array
    for i in range(data1.shape[0]):
        for j in range(data2.shape[0]):
            # Calculate Frobenius norm (Euclidean distance for matrices)
            distances[i, j] = np.linalg.norm(data1[i] - data2[j])

    # Solve the Optimal Transport problem
    optimal_transport_plan = ot.emd(weights1, weights2, distances)

    # Compute the Wasserstein distance (Earth Mover's Distance)
    wasserstein_distance = np.sum(optimal_transport_plan * distances)

    return wasserstein_distance
```

## Appendix D: Logits distance based on non-shuffled inputs

Logits distance obtained from non-shuffled inputs. The 1st table presents the distance calculated directly from the raw logits and input data that contain price scales; the 2nd table removes the price scale differences. CV (std/mean) measures data dispersion. A lower CV indicates the distance values are more consistent and less dispersed.

### 1st Table: Distances with Price Scale

| ticker | Volatility | Fully-trained Model | | | | Optimal Model | | | | No-training Model | | | | Input_Data |
|--------|-----------|----------|----------|-----------|-----------|----------|----------|----------|-----------|----------|----------|-----------|-----------|----------|
| | | Standard | DiffNeuron | LessLayer | MoreLayer | Standard | DiffNeuron | LessLayer | MoreLayer | Standard | DiffNeuron | LessLayer | MoreLayer | Distance |
| CET | 1.2% | 21.4 | 18.2 | 21.7 | 221.0 | 21.8 | 17.7 | 43.0 | 383.4 | 238.1 | 46.6 | 144.5 | 31.2 | 32.1 |
| AJG | 1.4% | 212.1 | 172.2 | 207.7 | 2,143.0 | 215.5 | 174.0 | 397.8 | 3,583.5 | 916.3 | 398.1 | 959.7 | 269.2 | 340.0 |
| FNWD | 1.5% | 23.8 | 19.9 | 23.1 | 251.1 | 23.6 | 20.0 | 47.1 | 413.2 | 214.6 | 46.8 | 128.1 | 30.4 | 39.1 |
| CFFN | 1.5% | 6.5 | 5.4 | 6.3 | 68.6 | 6.8 | 5.4 | 14.1 | 113.3 | 64.7 | 21.6 | 47.7 | 12.1 | 10.0 |
| RY | 1.5% | 64.3 | 53.7 | 63.8 | 667.2 | 68.2 | 54.2 | 128.5 | 1,113.7 | 488.3 | 171.2 | 477.1 | 117.1 | 97.2 |
| FDBC | 2.4% | 59.9 | 51.1 | 60.3 | 610.1 | 61.9 | 51.3 | 116.7 | 1,011.4 | 414.4 | 90.4 | 225.4 | 86.6 | 98.1 |
| VABK | 2.4% | 23.7 | 20.0 | 23.8 | 242.3 | 23.8 | 19.8 | 46.5 | 407.5 | 235.1 | 42.8 | 131.6 | 35.0 | 39.3 |
| FCF | 2.4% | 7.5 | 6.2 | 7.1 | 76.5 | 7.5 | 6.1 | 15.5 | 140.5 | 62.5 | 18.9 | 43.5 | 11.1 | 12.5 |
| PFC | 2.4% | 21.6 | 19.1 | 21.4 | 236.0 | 23.1 | 18.8 | 43.6 | 363.2 | 196.9 | 45.5 | 119.7 | 31.2 | 34.7 |
| BUSE | 2.4% | 40.8 | 35.3 | 42.3 | 426.4 | 42.5 | 35.3 | 83.7 | 729.1 | 236.4 | 49.9 | 139.4 | 40.3 | 78.1 |
| SLNH | 8.8% | 482.4 | 403.4 | 489.8 | 4,832.8 | 487.7 | 399.3 | 954.7 | 8,200.6 | 1,598.5 | 506.3 | 1,233.3 | 389.7 | 899.5 |
| CARV | 9.0% | 279.2 | 234.8 | 285.8 | 2,976.0 | 287.9 | 232.5 | 581.6 | 4,836.8 | 785.5 | 352.6 | 646.4 | 276.7 | 469.1 |
| HGBL | 13.9% | 2.4 | 1.6 | 2.6 | 17.9 | 2.6 | 1.8 | 4.7 | 52.1 | 5.2 | 0.3 | 3.1 | 2.7 | 3.9 |
| KINS | 16.5% | 5.5 | 4.7 | 5.7 | 54.6 | 5.7 | 4.8 | 10.9 | 96.5 | 35.5 | 9.1 | 22.2 | 6.4 | 10.8 |
| WT | 49.8% | 10.9 | 7.3 | 9.2 | 84.5 | 9.1 | 7.4 | 17.2 | 146.0 | 49.8 | 16.6 | 24.7 | 8.3 | 17.0 |
| mean | 7.8% | 84.1 | 70.2 | 84.7 | 860.5 | 85.9 | 69.9 | 167.0 | 1,439.4 | 369.5 | 121.1 | 289.8 | 89.9 | 145.4 |
| std | 12.6% | 136.6 | 114.0 | 138.6 | 1,387.3 | 138.7 | 113.1 | 272.0 | 2,328.2 | 434.2 | 162.2 | 375.4 | 121.6 | 247.7 |
| std/mean | 162.0% | 1.62 | 1.62 | 1.64 | 1.61 | 1.62 | 1.62 | 1.63 | 1.62 | 1.18 | 1.34 | 1.30 | 1.35 | 1.70 |

### 2nd Table: Distances without Price Scale

| ticker | Volatility | Fully-trained Model | | | | Optimal Model | | | | No-training Model | | | | Input_Data |
|--------|-----------|----------|----------|-----------|-----------|----------|----------|----------|-----------|----------|----------|-----------|-----------|----------|
| | | Standard | DiffNeuron | LessLayer | MoreLayer | Standard | DiffNeuron | LessLayer | MoreLayer | Standard | DiffNeuron | LessLayer | MoreLayer | Distance |
| CET | 1.2% | 0.7 | 0.6 | 0.7 | 6.9 | 0.7 | 0.6 | 1.3 | 11.9 | 7.4 | 1.5 | 4.5 | 1.0 | 1.0 |
| AJG | 1.4% | 0.6 | 0.5 | 0.6 | 6.3 | 0.6 | 0.5 | 1.2 | 10.5 | 2.7 | 1.2 | 2.8 | 0.8 | 1.0 |
| FNWD | 1.5% | 0.6 | 0.5 | 0.6 | 6.4 | 0.6 | 0.5 | 1.2 | 10.6 | 5.5 | 1.2 | 3.3 | 0.8 | 1.0 |
| CFFN | 1.5% | 0.7 | 0.5 | 0.6 | 6.8 | 0.7 | 0.5 | 1.4 | 11.3 | 6.5 | 2.2 | 4.8 | 1.2 | 1.0 |
| RY | 1.5% | 0.7 | 0.6 | 0.7 | 6.9 | 0.7 | 0.6 | 1.3 | 11.5 | 5.0 | 1.8 | 4.9 | 1.2 | 1.0 |
| FDBC | 2.4% | 0.6 | 0.5 | 0.6 | 6.2 | 0.6 | 0.5 | 1.2 | 10.3 | 4.2 | 0.9 | 2.3 | 0.9 | 1.0 |
| VABK | 2.4% | 0.6 | 0.5 | 0.6 | 6.2 | 0.6 | 0.5 | 1.2 | 10.4 | 6.0 | 1.1 | 3.3 | 0.9 | 1.0 |
| FCF | 2.4% | 0.6 | 0.5 | 0.6 | 6.1 | 0.6 | 0.5 | 1.2 | 11.2 | 5.0 | 1.5 | 3.5 | 0.9 | 1.0 |
| PFC | 2.4% | 0.6 | 0.6 | 0.6 | 6.8 | 0.7 | 0.5 | 1.3 | 10.5 | 5.7 | 1.3 | 3.5 | 0.9 | 1.0 |
| BUSE | 2.4% | 0.5 | 0.5 | 0.5 | 5.5 | 0.5 | 0.5 | 1.1 | 9.3 | 3.0 | 0.6 | 1.8 | 0.5 | 1.0 |
| SLNH | 8.8% | 0.5 | 0.4 | 0.5 | 5.4 | 0.5 | 0.4 | 1.1 | 9.1 | 1.8 | 0.6 | 1.4 | 0.4 | 1.0 |
| CARV | 9.0% | 0.6 | 0.5 | 0.6 | 6.3 | 0.6 | 0.5 | 1.2 | 10.3 | 1.7 | 0.8 | 1.4 | 0.6 | 1.0 |
| HGBL | 13.9% | 0.6 | 0.4 | 0.7 | 4.5 | 0.7 | 0.5 | 1.2 | 13.2 | 1.3 | 0.1 | 0.8 | 0.7 | 1.0 |
| KINS | 16.5% | 0.5 | 0.4 | 0.5 | 5.1 | 0.5 | 0.4 | 1.0 | 8.9 | 3.3 | 0.8 | 2.1 | 0.6 | 1.0 |
| WT | 49.8% | 0.6 | 0.4 | 0.5 | 5.0 | 0.5 | 0.4 | 1.0 | 8.6 | 2.9 | 1.0 | 1.5 | 0.5 | 1.0 |
| mean | 7.8% | 0.6 | 0.5 | 0.6 | 6.0 | 0.6 | 0.5 | 1.2 | 10.5 | 4.1 | 1.1 | 2.8 | 0.8 | 1.0 |
| std | 12.6% | 0.0 | 0.0 | 0.0 | 0.8 | 0.1 | 0.0 | 0.1 | 1.2 | 1.9 | 0.5 | 1.3 | 0.2 | 0.0 |
| std/mean | 162.0% | 0.08 | 0.10 | 0.08 | 0.13 | 0.09 | 0.08 | 0.10 | 0.12 | 0.46 | 0.47 | 0.48 | 0.30 | 0.00 |

## Appendix E: Logits Distance Based on Shuffled Inputs

Logits distance obtained from shuffled inputs. The 1st table presents the distance calculated directly from the raw logits and input data that contain price scales; The 2nd table removes the price scale differences. CV (std/mean) measures data dispersion. A lower CV indicates that the distance values are more consistent and less dispersed.

### 1st Table: Distances with Price Scale

| ticker | Volatility | Fully-trained Model | | | | Optimal Model | | | | No-training Model | | | | Input_Data |
|---|---|---|---|---|---|---|---|---|---|---|---|---|---|---|
| | | Standard | DiffNeuron | LessLayer | MoreLayer | Standard | DiffNeuron | LessLayer | MoreLayer | Standard | DiffNeuron | LessLayer | MoreLayer | Distance |
| CET | 1.2% | 56.1 | 13.7 | 22.3 | 10.4 | 87.9 | 14.7 | 25.6 | 18.7 | 43.8 | 29.4 | 204.5 | 18.0 | 32.1 |
| AJG | 1.4% | 524.4 | 121.2 | 206.3 | 92.3 | 779.6 | 132.9 | 231.8 | 142.8 | 135.0 | 151.1 | 1,059.1 | 126.1 | 340.0 |
| FNWD | 1.5% | 68.6 | 16.2 | 27.0 | 12.1 | 101.5 | 17.9 | 30.9 | 24.7 | 26.3 | 28.3 | 207.9 | 33.9 | 39.1 |
| CFFN | 1.5% | 18.2 | 4.5 | 6.9 | 3.3 | 26.5 | 4.9 | 8.1 | 9.0 | 2.8 | 8.9 | 81.9 | 3.4 | 10.0 |
| RY | 1.5% | 174.6 | 41.0 | 69.8 | 32.2 | 260.9 | 45.9 | 77.7 | 61.1 | 60.2 | 74.4 | 484.3 | 55.3 | 97.2 |
| FDBC | 2.4% | 169.6 | 41.2 | 66.5 | 30.7 | 256.8 | 44.3 | 77.5 | 54.2 | 55.4 | 45.0 | 319.5 | 52.0 | 98.1 |
| VABK | 2.4% | 68.6 | 16.3 | 26.7 | 12.2 | 103.1 | 17.8 | 29.9 | 25.2 | 35.7 | 29.3 | 197.4 | 20.6 | 39.3 |
| FCF | 2.4% | 21.9 | 5.1 | 8.4 | 3.9 | 32.5 | 5.6 | 9.1 | 9.3 | 3.3 | 7.5 | 76.1 | 4.2 | 12.5 |
| PFC | 2.4% | 62.5 | 15.3 | 25.1 | 11.6 | 95.8 | 16.7 | 28.5 | 22.3 | 30.7 | 25.1 | 144.0 | 7.3 | 34.7 |
| BUSE | 2.4% | 119.8 | 28.9 | 49.3 | 21.9 | 180.1 | 32.3 | 55.7 | 25.6 | 22.6 | 26.8 | 178.0 | 27.9 | 78.1 |
| SLNH | 8.8% | 1,547.0 | 329.9 | 572.0 | 250.4 | 2,059.4 | 373.0 | 639.8 | 355.2 | 180.4 | 256.2 | 1,905.4 | 125.0 | 899.5 |
| CARV | 9.0% | 719.8 | 179.0 | 306.1 | 137.6 | 1,156.0 | 198.4 | 338.3 | 240.4 | 130.3 | 134.3 | 1,026.9 | 79.9 | 469.1 |
| HGBL | 13.9% | 5.4 | 1.4 | 3.0 | 1.2 | 8.2 | 1.9 | 3.6 | 2.7 | 0.1 | 0.1 | 9.9 | 0.1 | 3.9 |
| KINS | 16.5% | 16.1 | 3.9 | 6.7 | 3.0 | 24.5 | 4.5 | 7.6 | 6.0 | 1.3 | 3.4 | 36.1 | 1.7 | 10.8 |
| WT | 49.8% | 25.2 | 5.9 | 10.7 | 4.6 | 38.3 | 7.1 | 12.1 | 9.1 | 2.7 | 3.5 | 41.9 | 1.8 | 17.0 |
| mean | 7.8% | 239.9 | 54.9 | 93.8 | 41.8 | 347.4 | 61.2 | 105.1 | 67.1 | 48.7 | 54.9 | 398.2 | 37.1 | 145.4 |
| std | 12.6% | 415.4 | 90.9 | 157.2 | 69.2 | 572.2 | 102.2 | 175.4 | 102.3 | 56.2 | 72.2 | 531.6 | 42.9 | 247.7 |
| std/mean | 162.0% | 1.73 | 1.66 | 1.68 | 1.65 | 1.65 | 1.67 | 1.67 | 1.53 | 1.15 | 1.31 | 1.34 | 1.16 | 1.70 |

### 2nd Table: Distances without Price Scale

| ticker | Volatility | Fully-trained Model | | | | Optimal Model | | | | No-training Model | | | | Input_Data |
|---|---|---|---|---|---|---|---|---|---|---|---|---|---|---|
| | | Standard | DiffNeuron | LessLayer | MoreLayer | Standard | DiffNeuron | LessLayer | MoreLayer | Standard | DiffNeuron | LessLayer | MoreLayer | Distance |
| CET | 1.2% | 1.7 | 0.4 | 0.7 | 0.3 | 2.7 | 0.5 | 0.8 | 0.6 | 1.4 | 0.9 | 6.4 | 0.6 | 1.0 |
| AJG | 1.4% | 1.5 | 0.4 | 0.6 | 0.3 | 2.3 | 0.4 | 0.7 | 0.4 | 0.4 | 0.4 | 3.1 | 0.4 | 1.0 |
| FNWD | 1.5% | 1.8 | 0.4 | 0.7 | 0.3 | 2.6 | 0.5 | 0.8 | 0.6 | 0.7 | 0.7 | 5.3 | 0.9 | 1.0 |
| CFFN | 1.5% | 1.8 | 0.4 | 0.7 | 0.3 | 2.6 | 0.5 | 0.8 | 0.9 | 0.3 | 0.9 | 8.2 | 0.3 | 1.0 |
| RY | 1.5% | 1.8 | 0.4 | 0.7 | 0.3 | 2.7 | 0.5 | 0.8 | 0.6 | 0.6 | 0.8 | 5.0 | 0.6 | 1.0 |
| FDBC | 2.4% | 1.7 | 0.4 | 0.7 | 0.3 | 2.6 | 0.5 | 0.8 | 0.6 | 0.6 | 0.5 | 3.3 | 0.5 | 1.0 |
| VABK | 2.4% | 1.7 | 0.4 | 0.7 | 0.3 | 2.6 | 0.5 | 0.8 | 0.6 | 0.9 | 0.7 | 5.0 | 0.5 | 1.0 |
| FCF | 2.4% | 1.8 | 0.4 | 0.7 | 0.3 | 2.6 | 0.5 | 0.7 | 0.7 | 0.3 | 0.6 | 6.1 | 0.3 | 1.0 |
| PFC | 2.4% | 1.8 | 0.4 | 0.7 | 0.3 | 2.8 | 0.5 | 0.8 | 0.6 | 0.9 | 0.7 | 4.2 | 0.2 | 1.0 |
| BUSE | 2.4% | 1.5 | 0.4 | 0.6 | 0.3 | 2.3 | 0.4 | 0.7 | 0.3 | 0.3 | 0.3 | 2.3 | 0.4 | 1.0 |
| SLNH | 8.8% | 1.7 | 0.4 | 0.6 | 0.3 | 2.3 | 0.4 | 0.7 | 0.4 | 0.2 | 0.3 | 2.1 | 0.1 | 1.0 |
| CARV | 9.0% | 1.5 | 0.4 | 0.7 | 0.3 | 2.5 | 0.4 | 0.7 | 0.5 | 0.3 | 0.3 | 2.2 | 0.2 | 1.0 |
| HGBL | 13.9% | 1.4 | 0.4 | 0.8 | 0.3 | 2.1 | 0.5 | 0.9 | 0.7 | 0.0 | 0.0 | 2.5 | 0.0 | 1.0 |
| KINS | 16.5% | 1.5 | 0.4 | 0.6 | 0.3 | 2.3 | 0.4 | 0.7 | 0.6 | 0.1 | 0.3 | 3.3 | 0.2 | 1.0 |
| WT | 49.8% | 1.5 | 0.3 | 0.6 | 0.3 | 2.3 | 0.4 | 0.7 | 0.5 | 0.2 | 0.2 | 2.5 | 0.1 | 1.0 |
| mean | 7.8% | 1.7 | 0.4 | 0.7 | 0.3 | 2.5 | 0.4 | 0.8 | 0.6 | 0.5 | 0.5 | 4.1 | 0.3 | 1.0 |
| std | 12.6% | 0.1 | 0.0 | 0.0 | 0.0 | 0.2 | 0.0 | 0.1 | 0.1 | 0.4 | 0.3 | 1.8 | 0.2 | 0.0 |
| std/mean | 162.0% | 0.09 | 0.09 | 0.06 | 0.07 | 0.09 | 0.07 | 0.08 | 0.24 | 0.78 | 0.53 | 0.45 | 0.65 | 0.00 |

