# OpenReview forum: "Unveiling Transfer Learning Effectiveness Through Latent Feature Distributions"
_TMLR — Rejected by TMLR_

### Review · Reviewer_941F · 2025-09-20

**Summary Of Contributions:**

The paper studies the effects of pre-training, defined in the work as training a model on an initial, broader dataset (with train/val/test splits), on model performance when fine-tuned on a smaller held-out dataset (again with train/val/test splits). The paper claims to provide insights into why pre-training helps with transfer learning, such as showing consistency in learned representations under pre-training, and less variance in fine-tuning learning curves when some model parameters are initialized from pre-trained values. The paper also introduces a method of using Wasserstein distance to study latent representation evolution, and uses the method to reveal effects of pre-training on latent representations.

Strengths
* The experimental setting is clearly explained, from the domain (stock predictions), to how the data sequences are spit to create the transfer learning setting, to the model used (lstm), to experiment hyperparameters.
* Thorough ablation experiments covering numerous variations in the training and fine-tuning setup.
* Using Wasserstein distance to study latent representation evolution is new and appears to be effective.

Weaknesses
* Only a single domain (stock prediction) is studied, limiting the generality of the paper's claims.
* The transfer learning setting used in the paper is 15 stocks held-out from a larger set of stocks. The task (stock prediction) remains the same. This is not the standard transfer learning setting but closer to in-distribution generalization. This reduces the validity of the claims in the paper.
* The conclusions in the paper are somewhat trivial or well-known. For example, Table 2 shows that training only the prediction head with pre-trained "trunk" performs better than training a model with only the prediction head is not interesting. The result in Table 3, where fine-tuning the prediction head with pre-trained "trunk" performs better than fine-tuning the entire model from scratch, has been known since image classification using pre-trained ImageNet.

**Audience:**

No

**Audience Explanation:**

Given the very limited domain, low transfer setting, and triviality of the results, very few in the TMLR's audience would find the findings of this paper interesting or new. See discussion above for details.

**Claims And Evidence:**

No

**Claims Explanation:**

The claims in the submission are overly broad and not supported by convincing evidence. The main issue is the paper only studies a single setting of stock prediction. Hence it is difficult to see why the claims can be applied to general pre-training and transfer learning. Results on 2-3 more domains (such as image, text, etc.) would improve the claims.

In addition, the transfer learning setup studied in the paper is closer to in-distribution generalization than task transfer learning (see more details in "Weaknesses" above). There is no convincing evidence that the results apply to more drastic or task transfer learning settings.

**Requested Changes:**

[critical] Expand study to 2-3 more domains (image, text, etc.).

[critical] Define more standard transfer learning settings (such as to a new task, or more drastic shifts such as visual style in the image domain)

[strengthen] Experiment with newer models beyond LSTM, such as transformers.

---

### Review · Reviewer_kSo4 · 2025-09-29

**Summary Of Contributions:**

**Summary**

The paper investigates why transfer learning is effective by analyzing how pretrained models transform data in their latent feature space. The study focuses on stock market prediction with LSTM models and evaluates three hypotheses. The first examines whether pretrained weights provide stable and fast fine-tuning. The second tests whether pretrained layers produce more consistent features across different data subsets, using a Wasserstein distance–based method to track the evolution of logit distributions across frozen layers. The third extends this analysis to layer-wise consistency across the model.

**Strengths**

* The paper proposes Wasserstein distance as a practical, quantitative way to analyze representation evolution which can be potentially re-used in future analysis.
* The paper focuses not only on accuracy but also on stability/consistency, which is not often brought up in related arts.
* The use of the S&P 500 dataset in testing Hypothesis 3 is an interesting way to consider generalization.
* The paper is very explicit about the details of their experiments and analysis.

**Weakneses**

* The primary concern is the rather narrow scope of the paper. It focuses solely on stock market prediction and LSTM models, which limits the credibility of the claim that the findings are broadly applicable.
* The test of Hypothesis 1 feels incomplete. If the claim is "The pretrained weights in both frozen and unfrozen layers enable generalizable", why is the baseline not a randomly initialized LSTM of equal depth, instead of a simpler one-layer model?
* It would be nice to explain figure 5 better. Specifically, what drives the observed layer-wise unimodal dynamics?

**Audience:**

Yes

**Audience Explanation:**

Despite its narrow scope, the paper presents several findings that are interesting and worth further exploration. This includes
* The emphasis on feature consistency
* The use of Wasserstein distance for evaluating representation learning
* The observation in Figure 5 regarding layer-wise dynamics in terms of logit distances.

**Claims And Evidence:**

No

**Claims Explanation:**

Right now, the claim of providing a “unifying explanation” feels overstated, since the evaluation is limited to stock market prediction. It is difficult to assess whether the observations would hold beyond this specific setting.

**Requested Changes:**

It is difficult to propose major changes for this paper. As noted above, the central issue is the narrow experimental scope. However, asking for broader empirical evidence would essentially require switching to new datasets, and given the time-series nature of the data, this would also request changing the model architecture, which is too substantial a revision to reasonably expect.

A more feasible way to strengthen the paper could potentially be to explore different pretraining regimes within the current setup: for example, comparing large vs. small source datasets (in terms of pre-training data size), or related vs. unrelated sectors (e.g., agriculture vs. a randomly chosen unrelated sector). Such experiments would test whether the proposed Wasserstein method can truly distinguish between "good" and "bad" pretraining strategies, thus making the analysis more comprehensive.

Apart from this, the presentation of the paper could also be tightened, some details currently in the main text could be moved to the Appendix, such as the data preparation steps, the explanation of the LSTM model and associated regularization/normalization discussion, and the derivations of Equations (1–4).

Minor: Subsection 2.3 title "Hypotheses setupe" is a typo.

---

### Review · Reviewer_3nhG · 2025-10-18

**Summary Of Contributions:**

The paper investigates what makes Transfer Learning (TL) effective by hypothesizing that pretraining consistently transforms input distributions into generalizable internal representations. They use the Wasserstein distance to track layer-by-layer distributional changes in the latent feature space of an LSTM model for stock prediction. The study provides strong evidence that reusing pretrained weights leads to stable and efficient fine-tuning, supporting the idea that TL's success is due to this consistent feature transformation.

**Additional Comments:**

None

**Audience:**

Yes

**Audience Explanation:**

1. Using the Wasserstein Distance (WD) to measure feature consistency provides a new, objective, and scale-independent metric for analyzing transferability. This moves the discussion beyond qualitative concepts to a measurable statistical property of the latent space.

2. The work empirically validates the notion that pretraining ensures stability in fine-tuning, demonstrating how low Wasserstein Distance correlates with consistent, predictable performance across different runs, which is a critical finding for large-scale production models.

**Claims And Evidence:**

No

**Claims Explanation:**

The abstract and introduction claim to provide a "unifying explanation" for Transfer Learning (TL) effectiveness, explicitly referencing major domains like Large Language Models (LLMs), Computer Vision, and multimodal systems. However, the entire empirical study is restricted to a seven-layer LSTM model applied only to financial time-series data. This narrow scope is insufficient to support the paper's broad, universal claim.

**Requested Changes:**

1. The current claims do not align with the evaluation setup. The authors should either clearly limit their conclusions to the LSTM experiments on time-series stock market data or expand their study to include transfer learning with VLMs and LLMs. This clarification or broader validation is needed to ensure the paper’s claims are properly supported.

2. I think it could add meaningful depth to the paper to connect the idea of layer plasticity with the observed Wasserstein Distance (WD) patterns. Specifically, could the Wasserstein Distance be used as a proxy to study changes in plasticity? For example, identifying when a layer has passed its critical period and become less adaptable? Does a low Wasserstein Distance indicate an actual loss of plasticity (reflecting a deeper learning dynamic within the model), or is it primarily a statistical signal of representational stability? Clarifying this relationship would make it clearer whether the Wasserstein Distance captures underlying learning behavior or simply reflects surface-level consistency in features.

3. The stability (low WD) and volatility shown in Table 3 directly mirror the "warm start [1]" problem in Transfer Learning. The superiority of the random-head model (ft-Progressive) suggests the stable pretrained weights (ft-Unfrozen) introduce a rigidity (low plasticity) that prevents the search for a better optimum. The authors should discuss this explicitly. They should explore whether a successful but suboptimal warm start corresponds to a distinct pattern of WD in the frozen layers, clarifying the trade-off between stability and adaptability. This connection would transform their consistency metric into a powerful diagnostic tool for this problem.

A simple experiment can be to measure and study Wasserstein Distance Pattern. For the frozen layers, measure and compare the Wasserstein Distance between the pretrained feature distribution and the final fine-tuned feature distribution across these two contrasting cases and discuss rigidity plasticity tradeoff.

[1] Ash, J. and Adams, R.P., 2020. On warm-starting neural network training. Advances in neural information processing systems, 33, pp.3884-3894.

---

### Decision · Action_Editor_H35H · 2025-12-21

**Recommendation:** Reject

**Audience:**

Yes

**Audience Explanation:**

The topic of understanding transfer learning could be of interest to TMLR’s audience. On the other hand, the current paper examines only a single setting (stock market prediction) and it is unclear whether the claims can be generalized to broader pretraining and transfer learning scenarios.

**Claims And Evidence:**

No

**Claims Explanation:**

The paper aims to investigate why transfer learning is effective by analyzing how pretrained models transform data in their latent feature space. Unfortunately, all reviewers identified a major limitation that the experiments are restricted to a very specific setting: stock market prediction using LSTM models. Reviewers also pointed out other comments, but the authors did not provide a revision to address them during the rebuttal. Consequently, all reviewers recommended rejection of the current version.

After reading the paper, I agree with the reviewers’ assessment and also recommend rejection. The authors may consider addressing the reviewers’ comments to substantially strengthen the work. In addition, the manuscript would benefit from significant improvements in presentation and writing. For example, many equations in Section 2.3 appear redundant, and the underlying assumptions are not clearly stated. Table 6 is empty, and Figure 2 is low-resolution. Moreover, much recent work on understanding transfer learning from hidden representations is not discussed.